# Quercetin Reduces the Development of 2,3,7,8-Tetrachlorodibenzo-p-dioxin-Induced Cleft Palate in Mice by Suppressing CYP1A1 via the Aryl Hydrocarbon Receptor

**DOI:** 10.3390/nu14122448

**Published:** 2022-06-13

**Authors:** Keisuke Satake, Takenobu Ishii, Taiki Morikawa, Teruo Sakamoto, Yasushi Nishii

**Affiliations:** Department of Orthodontics, Tokyo Dental College, 2-9-18, Kandamisaki-Cho, Chiyoda-Ku, Tokyo 101-0061, Japan; satakekeisuke@tdc.ac.jp (K.S.); morikawataiki@tdc.ac.jp (T.M.); tesakamo@tdc.ac.jp (T.S.); nishii@tdc.ac.jp (Y.N.)

**Keywords:** quercetin, cleft palate, flavonoids, 2,3,7,8-tetrachlorodibenzo-p-dioxin, mouse embryonic palatal mesenchymal cells, aryl hydrocarbon receptor, AhR repressor, CYP1A1

## Abstract

Quercetin is a flavonoid with a wide range of pharmacological activities, including anticancer, antioxidant, and anti-inflammatory effects. Since it is a nutrient that can be consumed with a regular diet, quercetin has recently garnered interest. Quercetin acts as a phytochemical ligand for the aryl hydrocarbon receptor (AhR). Cleft lip and palate are among the most frequently diagnosed congenital diseases, and exposure to 2,3,7,8-tetrachlorodibenzo-p-dioxin (TCDD) during pregnancy induces cleft palate via AhR. In this study, we investigated the preventive effect of quercetin intake on the TCDD-induced cleft palate and its mechanism of action. The in vivo results suggest that quercetin intake by pregnant mice can prevent cleft palate in fetal mice. In vitro, the addition of TCDD induced a reduction in cell migration and the proliferation of mouse embryonic palatal mesenchymal cells, which was mitigated by the addition of quercetin. The addition of quercetin did not alter the mRNA expression levels of the AhR repressor but significantly suppressed mRNA expression of CYP1A1. In addition, the binding of AhR to a xenobiotic responsive element was inhibited by quercetin, based on a chemically activated luciferase expression assay. In conclusion, our results suggest that quercetin reduces the development of TCDD-induced cleft palate by inhibiting CYP1A1 through AhR.

## 1. Introduction

Quercetin is a flavonoid, and flavonoids demonstrate a wide range of pharmacological activities, including anticancer, antioxidant, and anti-inflammatory effects [1]. Recent epidemiological studies have demonstrated the protective effects of flavonoids against cardiovascular diseases, cancer, and various other chronic diseases, along with anti-obesity effects [2,3,4,5,6]. Quercetin is found in a variety of foods such as onions, apples, broccoli, and green tea, and it is therefore consumed daily as a part of the diet. The pharmacological activity of quercetin has garnered significant interest in recent years; and, in Japan, the National Agriculture and Food Research Organization (NARO), commissioned by the Ministry of Agriculture, Forestry and Fisheries (MAFF), has developed a variety of onion called “Quell Gold” with a high quercetin content. Flavonoids such as quercetin demonstrate various pharmacological activities by acting as phytochemical ligands for the aryl hydrocarbon receptor (AhR) [7,8]. However, the mechanisms associated with these direct impacts are unknown and need to be investigated.

Cleft lip and cleft palate are among the most frequently diagnosed congenital diseases, and they are attributed to genetic or environmental factors [9]. Cleft lip and palate can cause facial deformities and pronunciation problems that adversely affect patients and their families. For patients to lead a social life without feeling inconvenienced, various treatments are necessary over a long period of time, beginning immediately after birth and proceeding to adulthood. The treatment requires an extremely diverse approach that includes orthodontics, oral surgery, pediatric dentistry, prosthodontics, and all other areas of dentistry and medical-related departments as well as speech training by a speech pathologist and psychological counseling by a psychologist. Epidemiological studies have found a correlation between exposure to dioxins during pregnancy and an increased risk of a cleft lip and palate [10]. Experiments in mice have shown that exposure to 2,3,7,8-tetrachlorodibenzo-p-dioxin (TCDD) during organogenesis induces cleft palate [11]. AhR is reported to be involved in the development of TCDD-induced cleft palate because AhR knockout mice do not develop cleft palate when TCDD is administered to them [12]. TCDD is the most potent ligand for AhR; however, difficult metabolization of TCDD leads to persistent AhR activation and its dysregulation [13]. These factors are thought to be the causes of cleft palate development induced by TCDD. Various methods have been investigated for the prevention of cleft palate. As a simple and an effective way, prevention of cleft palate with nutrients obtained from the daily diet is being examined. Intake of folic acid and α-naphthoflavone reduces the incidence of TCDD-induced cleft palate in fetal mice [14].

These findings suggest that the pharmacological activity of quercetin may prevent the onset or reduce the severity of TCDD-induced cleft palate. However, no studies have investigated this hypothesis. Therefore, we aimed to investigate the preventive effect of quercetin intake on TCDD-induced cleft palate and elucidate its mechanism of action. Our results suggest that quercetin prevents the development of TCDD-induced cleft palate by inhibiting CYP1A1 through AhR.

## 2. Materials and Methods

### 2.1. Animals

A total of 51 healthy pregnant ICR mice (0 days of pregnancy) were obtained from Sankyo Lab Services (Tokyo, Japan). At Sankyo Lab Services, female and male mice were housed together in cages overnight, and the vaginal plugs were checked the next morning. The date of confirmation of the vaginal plug was gestational day 0 (GD0). Mice were housed in a controlled environment (24 ± 1 °C, 12 h light/dark cycle, ad libitum access to food and water). The experiment was approved by the Institutional Review Board of the Tokyo Dental College (approval number: 203114) and conducted in accordance with the guidelines for experimental animals specified by the college. After induction of anesthesia with an inhalant anesthetic (sevoflurance), the mice were euthanized by an intraperitoneal overdose of 150 mg/kg pentobarbital sodium, and samples were collected.

### 2.2. Chemicals

TCDD was purchased from AccuStandard (New Haven, CT, USA). Quercetin was purchased from Sigma-Aldrich (St. Louis, MO, USA). Olive oil, dimethyl sulfoxide (DMSO), and Dulbecco’s modified Eagle’s medium (DMEM) were purchased from Fujifilm Wako Pure Chemicals Co. (Tokyo, Japan).

### 2.3. Animal Treatment

#### 2.3.1. Confirmation of TCDD Concentration That Induced Cleft Palate

A total of 21 pregnant mice were randomly divided into 7 groups (*n* = 3 each). Seven concentration groups of TCDD were established, varying from 10 μg/kg to 40 μg/kg in 5 µg increments. At the GD12 stage, TCDD was diluted in olive oil to obtain corresponding solutions of TCDD with specific concentrations. These solutions were forcibly administered orally using a gastric tube. Pregnant mice were euthanized at the GD16 stage, and fetuses were collected. The number of surviving and dead fetuses was measured. Viable fetuses were washed in phosphate-buffered saline (PBS) and fixed overnight in 4% paraformaldehyde (PFA) for histological analysis. After fixation, the heads were dehydrated with ethanol, embedded in paraffin, and sectioned at a thickness of 4 µm. Sections of the palate were stained with hematoxylin and eosin (HE) and observed under a UPM Axio Phot2 microscope (Carl-Zeiss, Jena, Germany).

#### 2.3.2. Examination of the Dosage of Quercetin

Fifteen pregnant mice were randomly divided into five groups (*n* = 3 each). Three groups of quercetin doses were established: 0.02 mg per day, 0.09 mg per day, and 0.30 mg per day. Mice in group 1 were administered saline at the GD12 stage. Group 1 was used as a negative control group (Group 1: CTRL). The other four groups of mice received 25 µg/kg of TCDD diluted in olive oil at the GD12 stage, and the group of mice that received only TCDD at the GD12 stage was used as the positive control group (Group 2: TCDD). The remaining three groups received quercetin orally by gavage daily from GD1 to GD16 stages (Group 3: TCDD + Quercetin 0.02 mg, Group 4: TCDD + Quercetin 0.09 mg, Group 5: TCDD + Quercetin 0.30 mg). Pregnant mice were euthanized at the GD16 stage, and fetuses were collected. Sections of the palate were prepared as described in Section 2.3.1 and observed under a UPM Axio Phot2 microscope.

#### 2.3.3. Examination of the Interval of Quercetin Administration

Fifteen pregnant mice were randomly divided into five groups (*n* = 3 each). Olive oil was administered daily at predetermined intervals starting from the GD1 stage. Mice in group 1 were administered saline at the GD12 stage. Group 1 was used as a negative control group (Group 1: CTRL). The other four groups of mice received 25 µg/kg of TCDD diluted in olive oil at the GD12 stage. Mice that received only TCDD at the GD12 stage were used as the positive control group (Group 2: TCDD). There were three quercetin groups according to the interval of quercetin administration: daily, single-dose, and post-dose (group 3: daily dose, group 4: single dose, and group 5: the post-dose). The daily dose group received quercetin daily from GD1 to GD16. In the single-dose group, quercetin was administered simultaneously with TCDD at the GD12 stage. The post-dose group received quercetin only between GD14 and GD16 (Figure 1A). Quercetin was dissolved in olive oil and then administered. Pregnant mice were euthanized at the GD16 stage, and fetuses were collected. Sections of the palate were prepared as described in Section 2.3.1 and observed under a UPM Axio Phot2 microscope.

### 2.4. Cell Culture

Pregnant mice were euthanized at the GD12 stage, and the fetuses were collected. Only the palate mucosa of the fetal mice was selectively isolated under a microscope and digested with dispase II (Godo Shusei Co., Tokyo, Japan) and trypsin EDTA (Biological Industries, Beit Haemek, Israel) at 37 °C for 30 min. After centrifugation, mouse embryonic palatal mesenchymal (MEPM) cells were isolated using a cell strainer with a pore size of 70 µm (BD Falcon, Phoenix, AZ, USA). Cells were seeded in DMEM (DMEM, 10% FBS, 1% penicillin/streptomycin, 1% amphotericin B) and cultured for 3 days. The cells were maintained at 37 °C in an atmosphere containing 5% CO_2_. After confirming that the culture was 80% confluent, the cells were divided into four groups for further experiments and passaged. The breakdown of the four groups is presented as follows: 0.1% DMSO (Group 1: CTRL), 0.1% DMSO + 10 nM TCDD (Group 2: TCDD), 0.1% DMSO + 1 µM Quercetin (Group 3: Quercetin), and 0.1% DMSO + 10 nM TCDD + 1 µM Quercetin (Group 4: TCDD + Quercetin). A TCDD concentration of 10 nM has been established in previous studies as an effective concentration for AhR activation [15].

### 2.5. Cell Proliferation Assay

#### 2.5.1. Evaluation of Cell Migration Ability

Cell migration ability was evaluated using a scratch assay. Cells were plated in 6-well plates and cultured until they reached 100% confluence. At 100% confluency, the wells were scratched with a pipette tip to create grooves. The cells were washed twice with PBS to remove the cell debris. After scratching, the medium was replaced with the medium corresponding to each of the four groups, and the cells were cultured for 12 h. The width of the grooves in each group was measured immediately after the scratch (0 h) and after 12 h of incubation (12 h). The ratio (%) of 12 h groove width/0 h groove width was determined (*n* = 5).

#### 2.5.2. Evaluation of Cell Proliferative Capacity

Cell proliferative capacity was assessed using Click-iT EdU Imaging Kits (Invitrogen, Waltham, MA, USA). A cover glass was placed in a 24-well plate, and the cells (6 × 10^4^) were plated on the cover glass. The cells were cultured for 12 h in a medium corresponding to each of the four groups. After 12 h of incubation, EdU solution (final concentration 10 µM) was added to the culture medium, followed by incubation at 37 °C for 3 h. After treatment, the cells were fixed with 4% PFA for 15 min at room temperature and then treated with Triton X-100 (Nacalai Tesque Inc., Kyoto, Japan) for 5 min at room temperature to permeabilize the cell membrane. Next, EdU-positive cells were stained with a Click-iT reaction cocktail (30 min at room temperature), and nuclear staining was performed with Hoechst 33342 (Thermo Fisher Scientific, Waltham, MA, USA) (30 min at room temperature). The cells were observed using an LSM 880 microscope (Carl Zeiss) (*n* = 5).

### 2.6. Reverse Transcription-Quantitative Real-Time PCR (RT-qPCR) Analysis

RT-qPCR analysis was used to evaluate the mRNA expression levels of the AhR repressor (AhRR) and CYP1A1 after 24, 48, and 72 h of incubation. Total RNA was isolated using the Trizol reagent (Invitrogen), and the quantity and quality of the isolated mRNA were evaluated using a Nanodrop ND-100 spectrophotometer (Thermo Fisher Scientific). The mRNA was converted to cDNA using ReverTra Ace qPCR RT Master Mix with gDNA Remover (Toyobo Co., Osaka, Japan). For RT-qPCR, the reaction mixture was prepared using Thunderbird SYBR qPCR Mix (Toyobo), paired primers, and a defined amount of template cDNA. RT-qPCR reactions were performed using the primer sets listed in Table 1. RT-qPCR was performed using the 7500 Fast Real-Time PCR system (Applied Biosystems, Foster City, CA, USA) and 7500 Fast System SDS Software (Applied Biosystems). The initial denaturation was induced at 95 °C for 24 s, followed by 40 cycles of denaturation at 95 °C for 3 s, annealing at 60 °C for 5 s, and extension at 72 °C for 45 s. Relative expression ratios of markers were calculated using the double delta comparative threshold cycle method. The calculated values were normalized with that of the internal control (β-actin) (*n* = 5).

### 2.7. Chemically Activated Luciferase Expression (CALUX^®^) Assay

The binding ability of AhR to xenobiotic responsive element (XRE) was evaluated using a CALUX^®^ assay. Each sample solution (4 µL) was added to 400 µL of RPMI 1640 medium (RPMI 1640, 10% FBS, 1% penicillin/streptomycin) and agitated. The solutions were then added to mouse hepatocarcinoma cells H1L1 (1.5 × 10^5^ cells/well) in 96-well microplates, two wells at a time. The cells were then incubated in a CO_2_ incubator (37 °C, 5% CO_2_) for 24 h. After incubation, the medium was removed and 50 µL of luciferin (Bright Glo Luciferase Assay System; Promega, Madison, WI, USA) was added as a substrate, and the relative luminescence unit (RLU) was measured using a luminometer. Measurements were performed thrice on separate days, with two groups each at a time (n = 3). The CTRL group was 0.1% DMSO, the TCDD group was 0.1% DMSO + 1 nM TCDD, and the quercetin groups Q250, Q50, and Q25 corresponded to 250 µg/mL, 50 µg/mL, and 25 µg/mL of quercetin. The TCDD + quercetin groups (TCDD + Q250, TCDD + Q50, TCDD + Q25) were 0.1% DMSO + 1 nM TCDD + each concentration of quercetin.

### 2.8. EdU In Vivo Assay

Click-iT EdU Imaging Kits (Invitrogen, Waltham, MA, USA) were used to evaluate cell proliferative potential in vivo. Pregnant mice were injected intraperitoneally with EdU (100 mg/kg) daily from GD13 to GD15. Four groups (CTRL group, TCDD group, quercetin group, TCDD + quercetin group) were slaughtered at GD16 and fetuses were collected. Sections were prepared and stained for EdU-positive cells using Click-iT reaction cocktail, followed by nuclear staining using Hoechst 33342 (Thermo Fisher Scientific). Sections were observed using an LSM 880 microscope (Carl-Zeiss).

### 2.9. Immunohistochemistry of CYP1A1

Immunohistochemical staining was performed to detect the accumulation of the CYP1A1 protein in the palatal sections of mice at the GD16 stage. Antigen activation was performed using an Immunosaver (Nisshin-EM Co., Tokyo, Japan) at 98 °C for 45 min. To avoid nonspecific background staining, tissue sections were blocked with 1% bovine serum albumin (Sigma-Aldrich, St. Louis, MO, USA) for 60 min at room temperature. Sections were stained with CYP1A1 polyclonal antibody (13241-1-AP, Cosmobio, Tokyo, Japan; dilution 1:100) and IgG1 Isotype Control (MAB002, R&D Systems, MN, USA; dilution 1:200). The cells were stained at 4 °C overnight. The sections were then incubated with biotinylated secondary antibodies (Iwai Chemicals Co., Tokyo, Japan) for 2 h at room temperature. After incubation, streptavidin-horseradish peroxidase (BioLegend, San Diego, CA, USA) was applied for 60 min. The DAB Substrate Kit (Vector Laboratories, Burlingame, CA, USA) was used for color development according to the manufacturer’s instructions. The sections were then contrast-stained with hematoxylin. Immunohistochemical staining of sections of the palate was performed using a UPM Axio Phot2 (Carl-Zeiss).

### 2.10. Statistical Analysis

Data are presented as the mean ± SD derived from at least three independent experiments. First, we tested the normality of the incidence of cleft palate in fetal mice born to the same mother in each group. Differences between mean values were analyzed using Student’s t-test or one-way analysis of variance (ANOVA). Percentage data were analyzed using Pearson’s chi-square test or Fisher’s exact test. Statistical significance was set at *p* < 0.05. All statistical analyses were performed using SPSS (version 24.0; SPSS Inc., Chicago, IL, USA).

## 3. Results

### 3.1. TCDD Dosage Concentration

The presence of cleft palate was confirmed by gross findings and histological analysis (Figure 1B). As shown in Table 2, the incidence of cleft palate was 10.8% with a TCDD concentration of 10 µg/kg, 75.0% with a TCDD concentration of 15 µg/kg, 82.5% with a TCDD concentration of 20 µg/kg, 91.9% with a TCDD concentration of 25 µg/kg, 94.9% with a TCDD concentration of 30 µg/kg, 95.1% with a TCDD concentration of 35 µg/kg, and 97.3% with a TCDD concentration of 40 µg/kg. The incidence of cleft palate increased in a TCDD concentration-dependent manner. Also, embryonic lethality was observed in the groups exposed to TCDD concentrations above 30 µg/kg. Based on these results, the TCDD concentration of 25 µg/kg that induced the development of cleft palate but did not demonstrate embryonic lethality was considered as the dose concentration to be used in subsequent experiments.

### 3.2. Quercetin Dosage

The incidence of cleft palate in the TCDD group was 92.1%, whereas the incidence of cleft palate in the TCDD + Quercetin 0.02 mg group was 89.3%, and the incidence of cleft palate in the TCDD + Quercetin 0.09 mg group was 88.6%. There was no significant difference in the incidence of cleft palate between the TCDD + Quercetin 0.02 mg group and the TCDD + Quercetin 0.09 mg group. In the TCDD + Quercetin 0.30 mg group, the incidence of cleft palate was 70.3%; hence, a significant decrease was observed in the incidence of cleft palate compared to that noted in the TCDD group (*p* < 0.01) (Figure 1C). Based on these results, we decided to use a quercetin dose of 0.30 mg for the following experiments.

### 3.3. Timing of Quercetin Administration

The incidence of cleft palate was 0% in the CTRL group and 92.1% in the TCDD group. The incidence of cleft palate in the daily administration group was 68.9% and that in the single-dose group was 70.9%; hence, a significant decrease in the incidence of cleft palate was observed compared to that noted in the TCDD group (*p* < 0.01). The incidence of cleft palate in the post-dose group was 92.3%, and there was no significant difference in the incidence of cleft palate compared to that noted in the TCDD group. There was no significant difference in the incidence of cleft palate between the daily administration group and the single-dose group (Figure 1D).

### 3.4. Cell Migration Ability and Cell Proliferation Ability

Migration of cells toward the scratch region was observed from 0 h to 12 h. The ratio of the 12 h groove width/0 h groove width (%) was calculated, and the TCDD group showed a significantly higher value of 66.4% compared to the other three groups (*p* < 0.05). This result indicated that the addition of TCDD reduced the cell migration ability. There was no significant difference between the CTRL, quercetin, and TCDD + quercetin groups (Figure 2A,B).

The percentage of EdU-positive cells was significantly lower in the TCDD group than that in the CTRL group (*p* < 0.01). This result indicated that the addition of TCDD decreased cell proliferation. No significant differences were observed between the quercetin group and the CTRL group. Upon comparing the TCDD and quercetin groups, we observed that the percentage of EdU-positive cells was significantly lower in the TCDD group (*p* < 0.01). The percentage of EdU-positive cells was significantly lower in the TCDD group than that in the TCDD + Quercetin group (*p* < 0.05) (Figure 2C,D).

### 3.5. mRNA Expression of AhRR and CYP1A1

TCDD activates CYP1A1 gene expression via AhR. The AhRR gene is also a negative feedback mechanism for AhR. The expression level of AhRR was significantly increased in the TCDD and TCDD + Quercetin groups compared to that in the CTRL group (*p* < 0.01). However, there was no significant difference between the TCDD and the TCDD + Quercetin groups (Figure 3A). The expression of CYP1A1 was significantly higher in the TCDD group than that in the CTRL group (*p* < 0.01). No significant difference was observed between the TCDD + Quercetin group and the CTRL group. In comparison, there was a significant decrease in the expression of CYP1A1 in the TCDD + Quercetin group (*p* < 0.01) (Figure 3B).

### 3.6. Binding Ability of AhR and XRE

The binding ability of AhR to XRE was significantly higher in the TCDD group compared to that in the CTRL group (*p* < 0.01). No significant differences were observed between the Q250, Q50, and Q25 groups and the CTRL group. No significant differences were observed between the TCDD + Q250, TCDD + Q50, and TCDD + Q25 groups and the CTRL group (Figure 3C). Based on these results, we calculated the rate at which the binding ability was suppressed using the formula shown in Appendix A. The results of these calculations are shown in Figure 3D. The TCDD + Q250, TCDD + Q50, and TCDD + Q25 groups showed remarkable suppression, with suppression rates of 98.2%, 99.9%, and 100%, respectively. There was no significant difference in the suppression rates among the three groups.

### 3.7. Cell Proliferation in the Palate of Fetal Mice

White arrows indicate EdU-positive cells, where active cell proliferation is taking place. TCDD treatment suppresses cell proliferation and palatal process elongation (Figure 4A(b,f)). In the CTRL and the Quercetin groups, EdU-positive cells accumulate in the palatal processes (Figure 4A(a,c,e,g)). In the TCDD + Quercetin group, EdU-positive cells tend to accumulate in the center of the palatal process. In addition, the inhibition of cell proliferation by TCDD is rescued by quercetin treatment (Figure 4A(d,h)).

### 3.8. Accumulation of CYP1A1 in the Palate of Fetal Mice

In the CTRL group, the left and the right palatal processes were joined to form a normal palate. In the TCDD group, the left and the right palatal processes were completely separated, indicating a cleft palate. In addition, areas of high accumulation of CYP1A1 at the tip of the palatal process were identified symmetrically in both the left and the right palatal processes (Figure 4B(b,f)). The Quercetin group, as well as the CTRL group, showed normal palatal morphology. No CYP1A1 positive cells were observed in the CTRL and the Quercetin groups (Figure 4B(a,c,e,g)). The TCDD + Quercetin group showed normal palatal morphology as did the CTRL group. In the TCDD+Quercetin group, the palatal processes were fused, but there was a diffuse scattering of CYP1A1 positive cells on the palate (Figure 4B(d,h)).

## 4. Discussion

The development of a cleft palate can be induced by genetic or environmental factors [9]. In mice, the lateral palatine process begins to grow vertically along the lateral border of the tongue between the GD12–14 stages and fuses in the midline by lifting and growing horizontally over the tongue from GD14–15 [16]. The palate is then formed by the complete union of the left and the right palatal processes at the GD16 stage [17]. There is a correlation between exposure to dioxin chemicals during pregnancy and an increased risk of the development of a cleft lip and palate [10], and experiments in mice have shown that exposure to TCDD during organogenesis induces cleft palate [11]. In our study, to confirm the optimal concentration of TCDD that induces the development of the cleft palate, we conducted a confirmatory experiment by administering varying TCDD concentrations from 10 µg/kg to 40 µg/kg, with an increment of 5 µg. Based on the results, a TCDD concentration of 25 µg mg/kg was used for administration because it induced the development of cleft palate, but it did not demonstrate embryonic lethality. These results were consistent with previous studies on the creation of a mouse model of a cleft palate by TCDD administration [11,14].

AhR is involved in the development of the cleft palate attributed to TCDD [12]. Previous studies have suggested the efficacy of vitamins and AhR antagonists in preventing the development of a cleft palate [18,19]. Folic acid, a vitamin B, has been found to exert a protective effect against the TCDD-induced cleft palate. A previous study showed that folic acid (5 mg/kg) significantly reduces but does not completely prevent the incidence of TCDD-induced cleft palate in fetal mice [18]. The AhR antagonist α-naphthoflavon has also been reported to exert a preventive effect on the TCDD-induced cleft palate. This study showed that α-naphthoflavon significantly reduced the incidence of TCDD-induced cleft palate in fetal mice at both a single dose in GD12 and multiple doses in GD 8-13. However, even with the use of α-naphthoflavon, TCDD-induced cleft palate could not be completely prevented [19]. In a study comparing the preventive efficacy rates of folic acid and α-naphthoflavon, both substances significantly prevented TCDD-induced cleft palate, but there was no significant difference in prevention rates between the two groups [14]. In our study, we investigated the preventive effect of quercetin, a type of flavonoid, on the TCDD-induced cleft palate. Quercetin has a short half-life of approximately 11–24 h, making it difficult to add to water-soluble foods. In this study, we dissolved it in olive oil to make it fat-soluble and maintain its stable state as long as possible [20]. Three groups were established that received quercetin doses of either 0.02 mg per day, 0.09 mg per day, or 0.30 mg per day. These doses were determined based on the following criteria: the estimated daily mean intake of quercetin in Japan is 16.2 mg and the daily median intake is 15.5 mg [21]. Using this as a reference, we converted the quantity based on the weight of the mice and decided to administer 0.02 mg of quercetin per day. In addition, the quercetin-rich onion “Quell Gold,” which was developed by NARO in Japan, contains 70 mg of quercetin per 100 g of the edible part. This was converted based on the weight of the mice to establish the dosage of 0.09 mg quercetin per day. We also decided to administer 0.30 mg of quercetin per day based on a previous study comparing the effects of folic acid and α-naphthoflavon on TCDD-induced cleft palate [14]. The results showed that 0.30 mg of quercetin significantly reduced the incidence of TCDD-induced cleft palate. When the preventive effect of quercetin administration was examined at different intervals, a significant decrease in the incidence of TCDD-induced cleft palate was observed in the group that received quercetin daily from GD1 to GD16 and in the group that received a single dose only at GD12. There was no significant difference in the incidence of cleft palate between the two groups. The onset of TCDD-induced cleft palate was not prevented in the group that was administered quercetin later in the GD15-16 stage. These findings suggest that quercetin must be present in utero when the lateral palatal process begins palatogenesis to prevent cleft palate. It was suggested that quercetin antagonizes TCDD and binds to AhR, thereby contributing to the prevention of TCDD-induced cleft palate.

As mentioned earlier, AhR is thought to be involved in the development of TCDD-induced cleft palate, since AhR knockout mice do not develop cleft palate when TCDD is administered to them [12]. AhR is present in the cytoplasm bound to two molecules of heat shock protein of 90 kDa (HSP90), X-associated protein 2 (XAP2), and the 23 kDa co-chaperone protein (p23) [7]. XAP2 and p23 have been shown to stabilize the binding of AhR to HSP90 and maintain it in the cytoplasm [22]. HSP90 is essential for the activation of AhR, and studies using yeast as an expression system have reported that when HSP90 activity is deficient, AhR becomes unstable, and it does not induce transcriptional activity [23]. Since TCDD and quercetin are lipophilic, they can easily permeate the cell membrane and bind to AhR as its ligands in the cytoplasm. HSP90, a molecular chaperone, is thought to be necessary for AhR to maintain a conformation that allows it to bind to its ligands, and the molecule that binds to its ligand undergoes a conformational change and enters the nucleus in a complex with HSP90 and other molecules [24]. Upon entry into the nucleus, the AhR complex dissociates HSP90, XAP2, and p23 to form a heterodimer with aryl hydrocarbon receptor nuclear translocator (ARNT). This heterodimer of AhR and ARNT binds to a specific DNA sequence called XRE, which activates the expression of various genes, including CYP1A1 [25]. In addition, AhRR has been reported to be involved in a negative feedback mechanism of AhR. AhRR forms a heterodimer with ARNT. Therefore, AhRR represses AhR transcriptional activity by competitively binding to ARNT. In addition, since the amino acid sequence on the C-terminal side of AhRR is completely different from that of AhR, the binding of the AhRR-ARNT heterodimer to XRE does not result in the activation of downstream genes [26] (Figure 5A).

In our study, MEPM cells showed decreased cell migration and cell proliferation upon the addition of TCDD. The reduction in cell migration and proliferation capacity induced by TCDD has been reported in other studies, which is consistent with our results [27]. The TCDD + Quercetin group showed significantly improved cell migration and proliferation abilities compared to the TCDD group. There was no significant difference in cell migration and proliferation ability between the CTRL and the quercetin groups. These results suggest that the addition of quercetin to TCDD ameliorates cell migration and proliferation not because quercetin enhances the cell migration and proliferation rate but because quercetin prevents the decrease in cell migration and proliferation induced by TCDD. In a study that showed that TGF-β3 prevented TCDD-induced reduction in cell migration and proliferation, the increase in cell migration and proliferation by the addition of TGF-β3 was considered a factor in mediating its protective effect [27]. However, quercetin prevented the TCDD-induced decrease in cell migration and proliferation by a mechanism of action different from that of TGF-β3. Migration of highly motile medial edge epithelium (MEE) is also said to be involved in secondary palatal fusion [28]. In this study, MEPM cells containing MEE were harvested from the mouse palate to demonstrate the effects of TCDD and quercetin in vitro. However, we plan to study the migration ability in vivo in the future.

To investigate the mechanism of action of quercetin in more detail, we investigated the mRNA expression levels of AhRR and CYP1A1. AhRR is involved in a negative feedback mechanism used to regulate AhR activation. Therefore, an increase in AhRR expression levels indirectly suppresses CYP1A1 expression levels. In our study, AhRR mRNA expression levels were not significantly different between the TCDD and the TCDD + Quercetin groups. However, the mRNA expression level of CYP1A1 was significantly increased in the TCDD group compared to that in the CTRL group, and there was no significant difference between the CTRL and the TCDD + Quercetin groups. In addition, in a comparison between the TCDD group and the TCDD + Quercetin group, there was a significant decrease in the mRNA expression level of CYP1A1 in the TCDD + Quercetin group. These results indicate that quercetin suppresses the mRNA expression of CYP1A1. These results suggest that the protective effects of quercetin on cell migration and proliferation are not due to the increased expression of AhRR but due to the decreased expression of CYP1A1 (Figure 5B,C).

The CALUX^®^ assay (Hiyoshi Co., Shiga, Japan) is one of the bioassays used for evaluating dioxins using a reporter gene assay. It has been recognized globally as an official method for measuring dioxin. The gene under the control of AhR is recombined with the gene that encodes luciferase, and when AhR binds to the ligand, the production of both CYP1A1 protein and luciferase protein is induced. The expression level of CYP1A1 was quantitatively determined by measuring the luminescence of luciferase. It is usually used to measure the concentration and total toxicity of dioxins in environmental samples, such as exhaust gas, ash, soil, and wastewater, and biological samples, such as fish and dairy products [29]. Using the CALUX^®^ assay, we quantitatively evaluated the inhibitory effect of quercetin on AhR and XRE binding. The assay showed that binding was significantly inhibited in the TCDD + Quercetin group compared to that in the TCDD group.

To determine whether reduced expression of CYP1A1 is associated with a reduced incidence of cleft palate in vivo, we performed the EdU assay and immunohistochemical staining for cell proliferation and CYP1A1 accumulation in the palate of mouse fetuses. In our study, cell proliferation was suppressed, and palatal process elongation was inhibited in the TCDD group. Both the CTRL and the Quercetin groups had an accumulation of EdU positive cells in the palatal process. This suggests that quercetin alone does not affect cell proliferation. The TCDD + Quercetin group also had normal palatal morphology with an accumulation of EdU positive cells in the palatal processes. These results suggest that quercetin acts in competition with TCDD to rescue the decreased cell proliferative capacity and to promote palatal process elongation. Immunohistochemical staining for CYP1A1 showed no CYP1A1 positive cells in the CTRL and the Quercetin groups. In the TCDD group, CYP1A1 was accumulated at the tip of the palatal process. In the TCDD + Quercetin group, the palatal processes were fused, but there was a diffuse scattering of CYP1A1 positive cells on the palate. These findings suggest that both TCDD and quercetin, when present, regulate CYP1A1 expression by acting competitively against AhR.

CYP1A1 is a metabolism-related enzyme regulated by AhR [30]. Various studies have suggested that CYP1A1 overexpression is involved in the development of the cleft palate [31,32]. Overexpression of CYP1A1 mediated by AhR has been reported to inhibit cell proliferation and cause cell death [33]. These findings suggest that the TCDD-induced cleft palate can be attributed to the overexpression of CYP1A1, which impairs cell migration and proliferation in the palatine process. Our study showed that quercetin acts as an AhR agonist and suppresses CYP1A1 via AhR. According to one report, this can be explained by quercetin acting as an agonist to AhR but producing a different signal than TCDD. This report states that resveratrol represses CYP1A1 transcription by inhibiting AhR binding to XRE [34]. Quercetin, like resveratrol, is thought to inhibit CYP1A1 by blocking AhR binding to XRE. However, the activation of genes related to drug metabolism, such as CYP1A1, alone cannot fully explain the development of malformations such as cleft palate. In addition, even though quercetin can markedly suppress the AhR activity of TCDD in vitro, it does not completely prevent cleft palate in vivo. Further elucidation is needed to determine whether activation of genes other than CYP1A1 is involved in cleft palate, or whether it is due to an unknown pathway other than transcriptional activation mediated by AhR.

## 5. Conclusions

In this study, we investigated the preventive effect of quercetin intake on the TCDD-induced cleft palate and its mechanism of action. Our data show that quercetin intake by pregnant mice during palatogenesis can reduce the development of TCDD-induced cleft palate in fetal mice. In vitro, the addition of TCDD caused a decrease in cell migration and cell proliferation of MEPM cells, but these were protected by the addition of quercetin. In conclusion, quercetin reduces the development of TCDD-induced cleft palate in mice by inhibiting CYP1A1 via AhR.

However, despite the fact that CYP1A1 protein expression is largely suppressed in vitro, it does not completely prevent TCDD-induced cleft palate in vivo. This suggests that factors other than CYP1A1 may also be associated with the development of cleft palate induced by TCDD. Therefore, more detailed studies need to be conducted in the future to investigate the effects of TCDD and quercetin on other factors.

## Figures and Tables

**Figure 1 nutrients-14-02448-f001:**
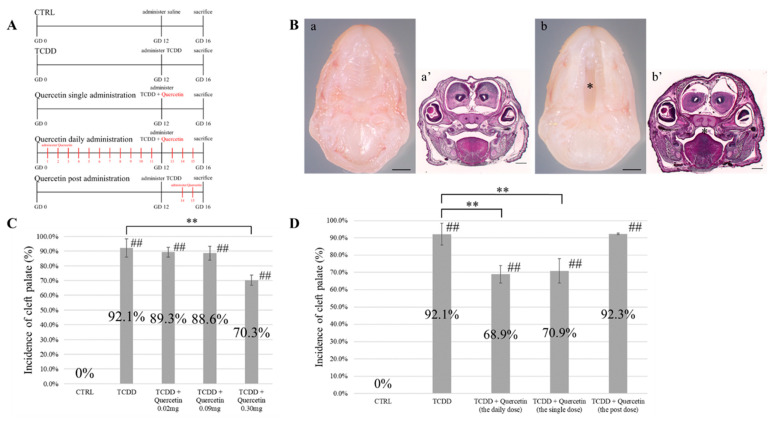
(**A**) Schematic diagram of TCDD and quercetin dosing schedule for control (CTRL) group, 2,3,7,8-tetrachlorodibenzo-p-dioxin (TCDD) group, Quercetin (daily administration) + TCDD group, Quercetin (single administration) + TCDD group, and Quercetin (post-administration) + TCDD group. Mice were euthanized on gestational day 16 (GD16). (**B**) Gross findings (a) and histological sections (a’) of a normal palate. Gross findings (b) and histological sections (b’) of a cleft palate. Scale bars = 500 µm. *: cleft palate area. (**C**) Incidence of cleft palate in the CTRL group, TCDD group, TCDD + Quercetin 0.02 mg group, TCDD + Quercetin 0.09 mg group, and TCDD + Quercetin 0.30 mg group. ** p* < 0.05; ** *p* < 0.01 vs. TCDD. ## *p* < 0.01 vs. CTRL. (**D**) Incidence of cleft palate in the CTRL group, TCDD group, TCDD + Quercetin (daily administration) group, TCDD + Quercetin (single administration) group, and TCDD + Quercetin (post-administration) group. * *p* < 0.05; ** *p* < 0.01 vs. TCDD. ## *p* < 0.01 vs. CTRL.

**Figure 2 nutrients-14-02448-f002:**
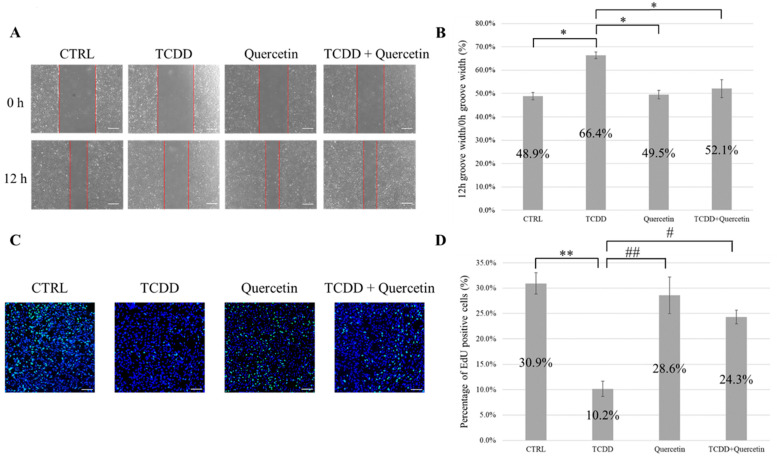
(**A**) Evaluation of the cell migration ability of mouse embryonic palatal mesenchymal (MEPM) cells by scratch assay. The cells of CTRL, TCDD, Quercetin, and TCDD + Quercetin groups were photographed immediately after scratching (0 h) and after 12 h of incubation (12 h). Scale bars = 500 µm. (**B**) The ratio of the width of the 12 h groove to the width of the 0 h groove for each group was determined. * *p* < 0.05; ** *p* < 0.01. (**C**) Evaluation of cell proliferative ability of MEPM cells as assessed by EdU staining. Blue: Hoechst, Green: EdU. Scale bars = 100 µm. (**D**) The percentage of EdU-positive cells in each group was determined. * *p* < 0.05; ** *p* < 0.01 vs. CTRL. *# p* < 0.05; *## p* < 0.01 vs. TCDD.

**Figure 3 nutrients-14-02448-f003:**
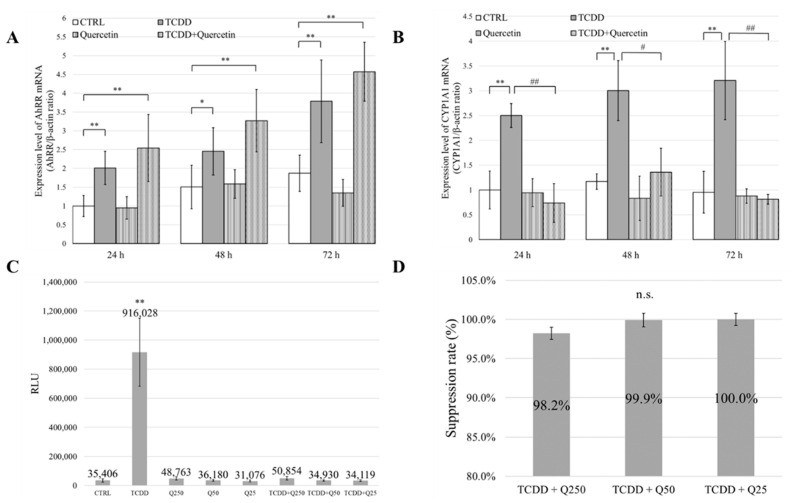
(**A**) Quantification of AhRR mRNA by RT-qPCR in mouse embryonic palatal mesenchymal (MEPM) cells at 24, 48, and 72 h. * *p* < 0.05; ** *p* < 0.01 vs CTRL. *# p* < 0.05; *## p* < 0.01 vs. TCDD. (**B**) Quantification of CYP1A1 mRNA by RT-qPCR in MEPM cells at 24, 48, and 72 h. * *p* < 0.05; ** *p* < 0.01 vs. CTRL. *# p* < 0.05; *## p* < 0.01 vs. TCDD. (**C**) Quantitative evaluation of the binding ability of AhR to XRE as determined by a chemically activated luciferase expression (CALUX^®^) assay. * *p* < 0.05; ** *p* < 0.01. (**D**) Rate at which the binding ability was suppressed by quercetin.

**Figure 4 nutrients-14-02448-f004:**
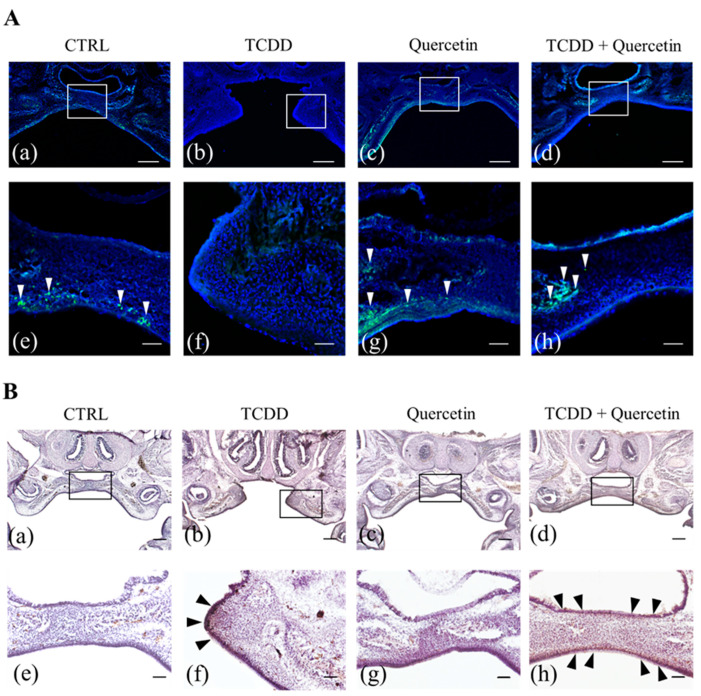
(**A**) Confirmation of cell proliferation in the palate of fetal mice by EdU staining. (a–d) are low magnification (scale bar = 200 µm). (e–h) are high magnification (scale bar = 50 µm). Nuclear (Blue: Hoechst 33342), EdU positive cells (Green: Alexa Fluor 488). White arrows indicate EdU positive cells. In the CTRL and Quercetin groups, EdU positive cells accumulate on the nasal and on the oral surfaces of the palatal process (a,c,e,g). In the TCDD group, EdU positive cells are not found in the palatal process at all, indicating that cell proliferation is suppressed by TCDD (b,f). In the TCDD + Quercetin group, EdU positive cells tend to accumulate in the center of the palatal process. It is also shown that the inhibition of cell proliferation by TCDD is rescued by quercetin (d,h). (**B**) Confirmation of CYP1A1 accumulation in the palate of fetal mice by immunohistochemistry staining. (a–d) are low magnification (scale bar = 100 µm). (e–h) are high magnification (scale bar = 75µm). Black arrows indicate CYP1A1 positive cells. No CYP1A1 positive cells are observed in the CTRL and the Quercetin groups (a,c,e,g). In the TCDD group, CYP1A1 is accumulated at the tip of the palatal process (b,f, black arrow). In the TCDD + Quercetin group, the palatal processes are fused, but there is a diffuse scattering of CYP1A1 positive cells on the palate (d,h, black arrows).

**Figure 5 nutrients-14-02448-f005:**
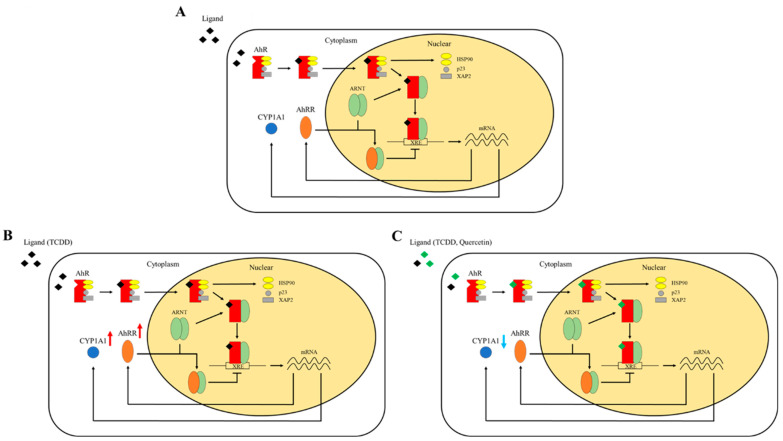
(**A**) The AhR pathway. AhR is present in the cytoplasm bound to two molecules of heat shock protein of 90 kDa (HSP90), X-associated protein 2 (XAP2), and the 23 kDa cochaperone protein (p23) [7]. Since TCDD and quercetin are lipophilic, they can easily permeate the cell membrane and bind to AhR as its ligands in the cytoplasm. AhR that binds to its ligand undergoes a conformational change and enters the nucleus in a complex with HSP90 and other molecules [24]. Upon entry into the nucleus, the AhR complex dissociates HSP90, XAP2, and p23 to form a heterodimer with aryl hydrocarbon receptor nuclear translocator (ARNT). This heterodimer of AhR and ARNT binds to xenobiotic responsive element (XRE), which activates the expression of various genes, including CYP1A1 [25]. AhRR has been reported to be involved in a negative feedback mechanism of AhR. AhRR forms a heterodimer with ARNT. Therefore, AhRR represses AhR transcriptional activity by competitively binding to ARNT [26]. (**B**) When TCDD binds to AhR as a ligand, both CYP1A1 and AhRR expression levels are significantly increased compared to the CTRL group. (**C**) When TCDD and quercetin bind to AhR as ligands, the expression level of AhRR is significantly increased compared to the CTRL group, but the expression level of CYP1A1 is not significantly different. Compared to the TCDD group, the expression level of AhRR is not significantly different, but the expression level of CYP1A1 is significantly decreased.

**Table 1 nutrients-14-02448-t001:** The RT-PCR primers.

Gene	Forward (5′–3′)	Reverse (3′–5′)
β-actin	CGGTTCCGATGCCCTGAGGCTCTT	CGTCACACTTCATGATGGAATTGA
CYP1A1	CTATCTGGGCTGTGGGCAA	CTGGCTCAAGCACAACTTGG
AhRR	GGAAGGCTGCTGTTGGAGTCTCT	TGGAAGCCCAGATAGTCCACGA

**Table 2 nutrients-14-02448-t002:** Incidence of cleft palate in each group.

Group	Number of Live Fetuses	Number of Viviparous Lethality	Number of Normal	Number of Cleft Palate	Incidence of Cleft Palate
TCDD 10 µg/kg	37	0	33	4	10.8%
TCDD 15 µg/kg	36	0	9	27	75.0%
TCDD 20 µg/kg	40	0	7	33	82.5%
TCDD 25 µg/kg	37	0	3	34	91.9%
TCDD 30 µg/kg	39	1	1	37	94.9%
TCDD 35 µg/kg	41	1	1	39	95.1%
TCDD 40 µg/kg	37	1	0	36	97.3%

## Data Availability

The data will be available on request.

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
