# Peer review of "Quercetin Reduces the Development of 2,3,7,8-Tetrachlorodibenzo-p-dioxin-Induced Cleft Palate in Mice by Suppressing CYP1A1 via the Aryl Hydrocarbon Receptor"

_nutrients, 2022, doi:10.3390/nu14122448_

Round 1

Reviewer 1 Report

Authors completely prove their research hypothesis with logical writing.

If references are corrected as journal format, it is good to publish.

Author Response

Thank you very much for providing important comments. We are thankful for the time and energy you expended. Our responses to the reviewers’ comments are as follow:

The format of the references was double-checked and corrected.

Reviewer 2 Report

Manuscript:ijms-1679736

Title: Quercetin Prevents the Development of 2,3,7,8-Tetrachlorodi- 2
benzo-p-dioxin-induced Cleft Palate in Mice by Suppressing 3 CYP1A1 via the Aryl Hydrocarbon Receptor

Authors: Keiuske Satake, Takenobu Ishii, Taiki Morikawa, Teruo Sakamoto, and Yasushi Nishii

The authors describe the introduction of quercetin into dioxin (TCDD) treated mice and show reduction of cleft palate incidence, which otherwise approaches 100% in TCDD exposed animals.

Overall, the manuscript is clearly written, well organized and appears to be scientifically sound. There are just a few relatively minor points that need to be addressed before the manuscript can be considered for publication. 

1. In the Title and Abstract, the authors state in several places that quercetin “PREVENTS” cleft palate. While this is technically correct, it leave the impression to the reader that it prevents clefting completely when in reality, incidence is reduced, albeit by a significant amount. There is still a 70.3% incidence of clefting, which is still high. 

- Instead, I would strongly suggest that the authors use “reduces incidence of” instead of “prevents”. This is much less misleading and more accurately describes their results. 

2. If the authors are going to argue for CYP1A1 as being the mechanism responsible for the reduction in cleft incidence, some discussion is warranted as to why reduction was only around 18-20%, with an incidence rate of 70.3% still remaining. 

3. The authors should consider increasing font size in the Figures. The graphs are particularly difficult to read

4. A pathway illustration similar to the one in Larigot et al., 2018 - Figure 3 would be very helpful as part of the discussion section. 

Author Response

Thank you very much for providing important comments. We are thankful for the time and energy you expended. Our responses to the reviewers’ comments are as follow:

1. Thank you for pointing this out. In order to accurately explain the result to the reader, we have changed the word "prevent" to "reduce".

2. Added to the discussion.

P13:496-500 " In addition, even though quercetin can markedly suppress the AhR activity of TCDD in vitro, it does not completely prevent cleft palate in vivo. Further elucidation is needed to determine whether activation of genes other than CYP1A1 is involved in cleft palate, or whether it is due to an unknown pathway other than transcriptional activation mediated by AhR."

3. Modified the font size of the figures to make them easier to read.

4. Thank you for your suggestion. We have added Figure 5 (AhR pathway diagram) to the Discussion section.

Reviewer 3 Report

This study aims to examine a preventative role of Quercetin in a TCDD-induced cleft palate mouse model. The authors found that Quercetin prevented TCDD-induced cleft palate through normalization of expression of Cyp1a1. However, there are several issues in current version of the manuscript.

  1. Luciferase assays were conducted in H1L1 cells, a mouse hepatocarcinoma cell line. It is unclear why the authors used this cell line. MEPM cells or alternatively O9-1 cells should be used in these experiments. Both cells are available for Luciferase assays.

  1. Quercetin was administered to mice as well as used for cell culture experiments; however, it is unclear how long Quercetin was active (chemical lifetime in vivo).

  1. BrdU incorporation assays should be conducted in vivo as well.

  1. Cell migration assays were conducted in cell culture. It is unclear how this cell characteristic contributes to palate formation. Please add experiments evaluating the cell migration in vivo.

  1. Section 3.5: It is unclear what the rationale of these experiments is. Especially, the relationship between AhRR and CYP1A1 should be described there, but not in the Discussion section.

  1. Figure 3: The relationship between AhRR and CYP1A1 is unclear. This may be co-incidence. There is no result directly supporting that TCDD-induced cleft palate was prevented by Quercetin through the AhR-mediated CYP1A1 inhibition.

  1. The in vivo rescue experiments should be analyzed in detail. For instance, the expression of Ahrr and Cyp1a1 should be evaluated there with qRT-PCR and IHC etc.

Author Response

Thank you very much for providing important comments. We are thankful for the time and energy you expended. Our responses to the reviewers’ comments are as follow:

1. Thank you for your suggestion. As you pointed out, it would be best to perform the experiment by introducing the luciferase gene into MEPM cells. However, in this study we are using the CALUX® assay principle to observe the ability of quercetin to bind AhR and XRE. Therefore, we believe that differences in cell lines are not a major issue.

In addition, the CALUX® assay is recognized worldwide as the official method for measuring dioxins, and mouse hepatocarcinoma cells H1L1 are used. It is a commonly used method for measuring dioxins and AhR activity.

2. Quercetin has a short half-life of approximately 11-24 hours, making it difficult to add to water-soluble foods. In this study, we dissolved it in olive oil to make it fat-soluble and maintain its stable state as long as possible. References added.(P11:381-383)

3. The EdU incorporation assays was also performed in vivo. Section 3.7 (Figure 4A)

4. In vivo cell proliferation was confirmed using the EdU uptake assay

5. Thank you for pointing this out. We have added a brief description of CYP1A1 and AhRR at the beginning of Section 3.5 to make it easier for the reader to understand.(P8:299-300)

6. As you indicated, there is a possibility of co-incidence between AhRR and CYP1A1. However, RT-qPCR results show that AhRR and CYP1A1 are both increased in the TCDD group compared to the CTRL group. On the other hand, AhRR is increased in the TCDD + Quercetin group compared to the CTRL group, but CYP1A1 is not significantly different. In addition, there is no significant difference in AhRR between the TCDD group and the TCDD + Quercetin group, but there is a significant difference in CYP1A1, and in IHC of CYP1A1, there is an accumulation of CYP1A1 in the TCDD group. These findings suggest that inhibition of CYP1A1 by quercetin contributes to the reduced incidence of TCDD-induced cleft palate.

7. Added IHC for CYP1A1. Section 3.8 (Figure 4B)

Round 2

Reviewer 3 Report

My previous comments (#4 and #7) are not adequately addressed. In addition, the quality of figures added to the revised manuscript is quite low and not acceptable (Figure 4, A and B).

3. BrdU incorporation assays should be conducted in vivo as well.

4. Cell migration assays were conducted in cell culture. It is unclear how this cell characteristic contributes to palate formation. Please add experiments evaluating the cell migration in vivo.

7. The in vivo rescue experiments should be analyzed in detail. For instance, the expression of Ahrr and Cyp1a1 should be evaluated there with qRT-PCR and IHC etc.

Author Response

We appreciate the time and effort you have dedicated to providing insightful feedback on ways to strengthen our paper. 

3. 

In vivo EdU incorporation assays were performed. (Figure. 4A)

The quality of the figures has also been revised to the highest possible quality, and the following information has been included in the article.

“White arrows indicate EdU-positive cells, where active cell proliferation is taking place. TCDD treatment suppresses cell proliferation and palatal process elongation (Figure 4A(b)(f)). In the CTRL and Quercetin groups, EdU positive cells accumulate in the palatal processes (Figure 4A(a)(c)(e)(g)). In the TCDD + Quercetin group, EdU positive cells tend to accumulate in the center of the palatal process. In addition, the inhibition of cell proliferation by TCDD is rescued by quercetin treatment (Figure 4A(d)(h)).” (P9:329-335)

“To determine whether reduced expression of CYP1A1 is associated with reduced incidence of cleft palate in vivo, we performed the EdU assay and immunohistochemical staining for cell proliferation and CYP1A1 accumulation in the palate of mouse fetuses. In our study, cell proliferation was suppressed, and palatal process elongation was inhibited in the TCDD group. Both the CTRL and Quercetin groups had an accumulation of EdU positive cells in the palatal process. This suggests that quercetin alone does not affect cell proliferation. The TCDD + Quercetin group also had normal palatal morphology with an accumulation of EdU positive cells in the palatal processes. These results suggest that quercetin acts in competition with TCDD to rescue the decreased cell proliferative capacity and promote palatal process elongation.” (P14:507-516)

4.

Thank you for pointing this out. We have included the following new information.

“Migration of highly motile medial edge epithelium (MEE) is also said to be involved in secondary palatal fusion [28]. In this study, MEPM cells containing MEE were harvested from the mouse palate to demonstrate the effects of TCDD and quercetin in Vitro. However, we plan to study the migration ability in Vivo in the future.” (P12:456-460)

[28] Aoyama, G.; Kurosaki, H.; Oka, A.; Nakatsugawa, K.; Yamamoto, S.; Sarper, S.E.; Usami, Y.; Toyosawa, S.; Inubushi, T.; Isogai, Y.; Yamashiro, T. Observation of Dynamic Cellular Migration of the Medial Edge Epithelium of the Palatal Shelf in vitro. Front. Physiol. 2019, 10, 698. DOI: 10.3389/fphys.2019.00698.

7.

The quality of the figures has also been revised to the highest possible quality, and the following information has been included in the article.

“In the CTRL group, the left and right palatal processes were joined to form a normal palate. In the TCDD group, the left and right palatal processes were completely separated, indicating a cleft palate. In addition, areas of high accumulation of CYP1A1 at the tip of the palatal process were identified symmetrically in both the left and right palatal processes (Figure 4B(b)(f)). The Quercetin group, as well as the CTRL group, showed normal palatal morphology. No CYP1A1 positive cells were observed in the CTRL and Quercetin groups (Figure 4B(a)(c)(e)(g)). The TCDD + Quercetin group showed normal palatal morphology as did the CTRL group. In the TCDD+Quercetin group, the palatal processes were fused, but there was a diffuse scattering of CYP1A1 positive cells on the palate (Figure 4B(d)(h)).” (P9:339-348)

“Immunohistochemical staining for CYP1A1 showed no CYP1A1 positive cells in the CTRL and Quercetin groups. In the TCDD group, CYP1A1 was accumulated at the tip of the palatal process. In the TCDD + Quercetin group, the palatal processes were fused, but there was a diffuse scattering of CYP1A1 positive cells on the palate. These findings suggest that both TCDD and quercetin, when present, regulate CYP1A1 expression by acting competitively against AhR.” (P14:516-521)